# STFormer: Spatial Temporal Spiking Transformer

## Abstract

Spiking Neural Networks (SNNs) are a type of neural network that is modelled based on the biological processes that occur within the brain. This paper introduces "STFormer", an innovative SNNs architecture tailored for robust Spatial-Temporal feature extraction. STFormer integrates a Spatial Core to capture spatial features and a Temporal Core to extract temporal dynamics. The Spiking Neural (SN) layer transforms these spatial insights into spike-encoded representations, preserving the foundational principles of SNNs. The Temporal Core excels in capturing intricate temporal features and introduces Relative Position Embedding (RPE) with residual concatenation for precise temporal analysis. Additionally, it incorporates the Spiking Guided Attention (SGA) module to enhance contextual understanding. The SGA module deploys attention mechanisms to effectively capture Spatial-Temporal patterns, enriching contextual awareness. Processed features undergo refinement through the Multi-Layer Perceptron (MLP). The data underwent evaluation from both neuromorphic and non-neuromorphic datasets and achieved state-of-the-art results. Specifically, we attained an accuracy of 96.37% for CIFAR10 and 81.45% for CIFAR100. On the neuronmorphic dataset, we reached an accuracy of 83.1% for CIFAR10-DVS and 98.61% for DVS128 Gesture. STFormer demonstrates exceptional Spatial-Temporal feature extraction capabilities, showcasing its potential in diverse computer vision applications.

## 1 Introduction

Inspired by the human brain, deep Artificial Neural Networks (ANNs) have garnered significant success, particularly in areas such as intelligent transport and precision agriculture. However, these accomplishments come at a considerable computational cost. ANNs consume approximately 12 times Yao et al. (2022a) more energy than the human brain, rendering high-energy models tough to deploy onto resource-constrained appliances, for example, smartphones and IoT devices. Utilising the brain's efficient computational paradigm to create low-energy neural networks on such platforms holds significant value.

Energy-efficient Spiking Neural Networks (SNNs) are essentially solutions. Structurally, SNNs resemble traditional ANNs. However, unlike ANNs, the data in SNNs is composed of 0s and 1s, where 0 indicates no activity, while 1 activates neurons to emit a spike. The data matrix in SNNs is sparse, with only certain neurons being activated. This is why SNNs are energy-efficient.

There are two methods of obtaining a SNN. The first involves training an ANN and then converting it to a SNN, called ANN2SNN Bu et al. (2022a); Cao et al. (2015); Li et al. (2021). It may need significant time-steps, resulting in enhanced latency. Another option is implementing direct training methods, such as STDP and STBP Wu et al. (2018). STDP is particularly effective in utilizing information in both temporal and spatial dimensions, while STBP employs a gradient descent method similar to that used in ANNs. Using non-differentiable step functions for gradient backpropagation can lead to gradient explosion. Thus, currently, some training methods Yao et al. (2023); Zhou et al. (2022) use proxy functions such as ATan and Sigmoid. As neural networks deepen, many neurons in the deeper layers become dormant, giving rise to the issue of vanishing gradients. To tackle this issue, the Multi-Level Firing (MLF) approach was introduced by Feng et al. (2022). This method efficiently decreases the number of inactive neurons by co-controlling multiple thresholds.

Additionally, Fang et al. (2021a) implements residual connections with spiking neurons in order to ensure deeper models for SNNs. Consequently, SNNs can handle deeper architectures similarly to ANNs.

However, a performance gap still exists between SNNs and traditional ANNs. To address the lack of accuracy, the attention mechanism has been incorporated into SNNs. Zhou et al. (2022) introduced SSA, which leverages the self-attention capability and biological properties of SNNs. However, during computation, there is an abundance of non-spike data which contradicts the biological features of SNNs. As a means of maintaining biological properties, multiple LIF neurons are added in spike-driven transformer Yao et al. (2023) and spikingformer Zhou et al. (2023a). From a theoretical standpoint, spikingformer-cml Zhou et al. (2023b) optimises the placement of LIF neurons, effectively overcoming the imprecision of gradient backpropagation.

To address this gap and propel the field of spiking neural computation forward, we introduce the Spatial-Temporal Spiking Transformer (STFormer). STFormer represents a pioneering fusion of SNNs with the Transformer architecture, aiming to capture the rich spatial-temporal patterns inherent in spike data. Building upon well-established techniques such as the STBP training scheme, proxy gradient functions for gradient backpropagation smoothing, and biologically inspired attention mechanisms, our model ventures beyond the limitations of prior approaches.

In particular, STFormer places paramount emphasis on both the temporal and spatial dimensions of spike data, surpassing the preceding models' prioritization. Within the Temporal Core, we introduce a 3D convolution module, enabling simultaneous processing of information across multiple time-steps in a highly parallelized fashion. This innovation allows dynamic weight computation to accommodate features at different timescales, enhancing our model's temporal modeling capabilities. Addressing the spatial dimension, we redefine the approach to spatial feature extraction. By incorporating larger convolutional kernels with position encoding, our model extends its receptive field, thereby enriching its perceptual and expressive power. Additionally, our model incorporates the innovative Spiking Guided Attention (SGA) mechanism, representing a substantial advancement in Spatial-Temporal information processing. SGA fundamentally alters the way we discern intricate Spatial-Temporal patterns within the data, breaking free from the constraints of traditional attention mechanisms. It leverages the rich Spatial-Temporal context provided by STFormer, thereby significantly enhancing the model's capacity to emphasize salient features and comprehensively analyze both spatial and temporal aspects of the input data.

In summary, STFormer, equipped with the SGA mechanism, represents a groundbreaking advancement in the realm of SNNs, ushering in a new era of spatial-temporal information processing. Our approach seeks to bridge the gap between biological neural systems and computational models, providing a powerful tool for applications requiring nuanced spatial-temporal understanding.

Our contributions are summarized below:

- ST-Core Module: Introduction of the ST-Core module, prioritizing spatial-temporal properties in spike data processing within SNNs. It enhances feature extraction and model accuracy, applicable to diverse tasks.
- SGA Module: Integration of the SGA module to elevate contextual understanding and feature extraction capabilities, addressing the model's robustness and improving its ability to prioritize spatial-temporal information.
- Comprehensive theoretical analysis of ST-Core's role in spike data processing, offering valuable insights into its impact, complemented by visualizations of spike data.
- Extensive experiments confirming the proposed architecture's superiority over state-of-the-art SNNs on neuromorphic and non-neuromorphic datasets, underlining its practical significance in advancing Spatial-Temporal data processing.

## 2 RELATED WORK

### 2.1 TRAINING METHODS OF SNNS

SNNs are renown as third-generation neural networks Maass (1997) due to their high biological plausibility, event-driven properties, and low power consumption on neuromorphic hardware. With

the related research work carried out in recent years, SNNs have been developed vigorously and become a potential competitor of ANNs.

The key distinction between SNNs and ANNs lies in the use of spiking neurons as the activation layer Wu et al. (2018); Fang et al. (2021b); Yao et al. (2022b); Feng et al. (2022); Fang et al. (2023). These spiking neurons making the model more easily interpreted in biological terms and better equipped to capture temporal data. Whereas ANNs benefit from well-established and effective gradient backpropagation methods during training, SNNs have two widely used training approaches: conversion from ANN to SNN, and direct training using a surrogate gradient. ANN-to-SNN Conversion Bu et al. (2022a); Li et al. (2021); Cao et al. (2015) involves replacing the ReLU activation layer in a trained ANN network with a spiking neuron and adjusting certain hyper-parameters to achieve a highly accurate SNN. However, this method is characterized by long conversion time-steps as well as constraints associated with the original ANN design. In addressing these challenges, Wu et al. (2018) proposes the STBP method, which realizes the direct training of SNNs by adopting the surrogate gradient method and achieves a high level of accuracy with a very small time-step.

As SNNs have a distinct network structure compared to ANNs, they require specific training strategies to achieve superior performance. Examples include enhancing the BatchNorm layer Duan et al. (2022); Zheng et al. (2021), and using temporal-efficient training Deng et al. (2022). Implementing these efficient techniques has led to significant results in various fields. For instance, object detection can be achieved through Spiking-Yolo Kim et al. (2020) and ems-yolo Su et al. (2023), while semantic segmentation can be accomplished using Deep Spiking-UNet Li et al. (2023). In terms of large language models, SpikingGPT Zhu et al. (2023) and SpikingBert Bal & Sengupta (2023) are preferable, and for generative models, SpikingGAN Kotariya & Ganguly (2022) is the way to go. SpikingGCN Zhu et al. (2022) and SpikingGAT Wang & Jiang (2022) are recommended for Graph Neural Networks. Furthermore, neuromorphic chips such as TrueNorth Merolla et al. (2014), Loihi Davies et al. (2018), and Tianjic Pei et al. (2019) are now available, making it increasingly likely that Spiking Neural Networks will be widely used in the near future.

## 2.2 TRANSFORMER WITH SNNS

Thanks to the well-established and effective network architectures in ANNs, SNNs can utilise them to construct high-performance models, such as Hu et al. (2021a); Fang et al. (2021a); Yao et al. (2022a); Hu et al. (2021b). The attention mechanism, currently the most efficient method in ANNs, has also been integrated into SNNs, including the implementation of the Transformer, its most classic network architecture. Spikformer Zhou et al. (2022) is the initial directly-trained Transformer within SNNs. It adopts a new spike-form self-attention named Spiking Self Attention (SSA). However, the current configuration of the Spikformer, which includes residual He et al. (2016) connections, still involves non-spike computation. Therefore, the Spike-Driven Yao et al. (2023) Transformer and Spikingformer Zhou et al. (2023a) present novel structures for preserving the spike computation. The issue of non-spike computation is resolved by Spikingformer and Spike-Driven Transformer through repositioning of spiking neurons, and by introducing Spike-Driven Self-Attention (SDSA), respectively. These spiking neural networks based on the attention mechanism enable Transformer to be implemented in SNNs, but they rarely consider the spike data. As for this, we introduce the STFormer to better concern the spike data.

## 3 METHOD

We introduce the STFormer, a novel fusion of the Transformer architecture with the Spatial-Temporal Core. This section will begin by providing a concise overview of spike neurons' working principles. Subsequently, we will delve into the comprehensive framework of the STFormer, followed by an in-depth exploration of the Spatial-Temporal Core and the Spiking Guided Attention (SGA) module. Finally, we will discuss the energy consumption aspect.

## 3.1 PRELIMINARIES

In SNNs, spike neurons control the release of spikes based on a threshold. In this paper we use LIF neurons, which work in the following way:

Figure 1: Framework of the model. Within our model framework, RGB image data serves as the initial input, undergoing conversion into spike-encoded format through spike neurons. Subsequently, the spike-encoded data flow into the Spatial-Temporal Core, responsible for intricate Spatial-Temporal feature extraction. Following this stage, we introduce the Spiking Guided Attention (SGA) module, noted for its remarkable energy efficiency and precise feature focus. Lastly, the data is directed to the Spatial-Temporal Multi-Layer Perceptron (ST-MLP) for comprehensive processing and feature refinement, empowering our model to effectively harness Spatial-Temporal information across diverse applications, all while maintaining exceptional energy efficiency.

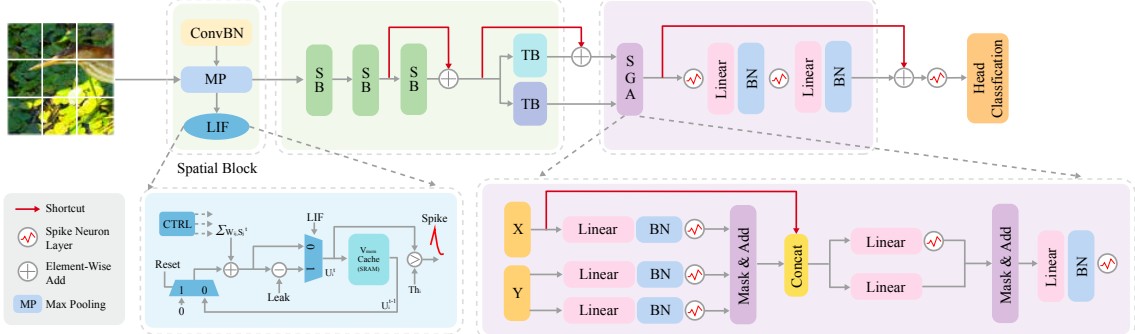

$$X_t^{L_j} = \sum_{i=1}^{N} w_{ij}^{L-1} o_i^{L-1} + b_j \tag{1}$$

$$O_t^{L-1} = HeavFunc(U_t^{L-1}, v_{th}) \tag{2}$$

$$U_t^L = k_\tau U_t^{L-1}(1 - O_t^{L-1}) + X_t^{L_j} V_{reset} \tag{3}$$

where $k_\tau$, $v_{th}$ and $V_{reset}$ are decay factor, firing threshold and reset membrane potential respectively, which are preset by default. $X_t^{L_j}$ denotes the input of the $j_{th}$ neuron of layer L, it can be calculated by last layers' output $o_i^{L-1}$ and weight matrix $w_{ij}^{L-1}$, $b_j$ denotes the bias. The $O^{L-1}$ is the output of the Heaviside step function, which is 0 or 1. After a spike activity, membrane potential $U_t$ will be reset by the Eq **??**.

## 3.2 OVERALL ARCHITECTURE

As illustrated in Figure 1, our architecture begins by processing the original data through the Spatial Core, where spatial features are meticulously extracted. In line with the commitment to preserving the biological characteristics inherent in Spiking Neural Networks (SNNs), we incorporate a Spiking Neural (SN) layer. This SN layer plays a pivotal role in transforming the output from the Spatial Core into a spike-encoded representation. These spike-encoded data are subsequently fed into the Temporal Core, where intricate temporal features are extracted.

To further enrich the temporal information and facilitate precise temporal analysis, we introduce the concept of Relative Position Embedding (RPE). RPE is applied to the output of the Temporal Core, enabling the incorporation of temporal positions for spike data. To ensure seamless integration and avoid information loss, we employ a residual concatenation strategy, combining the RPE-enhanced data with the original spike-encoded information.

The process can be summarized by the following equations:

$$x_S = SpatialCore(I), \qquad I \in \mathbb{R}^{T \times C \times H \times W}, x_S \in \mathbb{R}^{T \times N \times D} \tag{4}$$

$$S = SpikeNeuron(x_S), \qquad S \in \mathbb{R}^{T \times N \times D} \tag{5}$$

$$S_P = RPE(S) + x_S, \qquad S_P \in \mathbb{R}^{T \times N \times D} \tag{6}$$

$$\{S_{T1}, ..., S_{Tn}\} = TemporalCore(S_P), \qquad S_{Ti} \in \mathbb{R}^{T/n \times N \times D}, i \in \{1, ..., n\} \tag{7}$$

Here, $x_S$ represents the output from the Spatial Core, and $S_T$ denotes the output from the Temporal Core. This data processing section effectively combines spatial and temporal features, yielding data with rich Spatial-Temporal information.

Subsequently, the spatialtemporally enhanced features, now in spike-encoded form, are directed into the Spiking Guided Attention (SGA) module for attentive calculation. The results of this attention mechanism are then conveyed into the Multi-Layer Perceptron (MLP) for further processing. Following this stage, the data undergoes additional transformation within the spiking layer before proceeding to the Classification Header (CH), where classification decisions are made subsequent to Global Average Pooling (GAP).

$$S_{Ti}^{'} = SpikeNeuron(S_{Ti}), \qquad S_{Ti}^{'} \in \mathbb{R}^{T \times N \times D}, i \in \{1, ..., n\} \tag{8}$$

$$S_l = SGA(S_{T1}, ..., S_{Tn}), \qquad S_l \in \mathbb{R}^{T \times N \times D} \tag{9}$$

$$S_l^{'} = SpikeNeuron(S_l), \qquad S_l^{'} \in \mathbb{R}^{T \times N \times D} \tag{10}$$

$$S_T^{'} = ST - MLP(S_l^{'}) + S_l^{'}, \qquad S_T^{'} \in \mathbb{R}^{T \times N \times D} \tag{11}$$

$$Y = CH(GAP(S_T^{'})), \qquad Y \in \mathbb{R}^{T \times N \times D} \tag{12}$$

This comprehensive architectural framework capitalizes on the synergy between spatial and temporal information, effectively preserving the biological underpinnings of SNNs while enabling robust Spatial-Temporal feature extraction and classification.

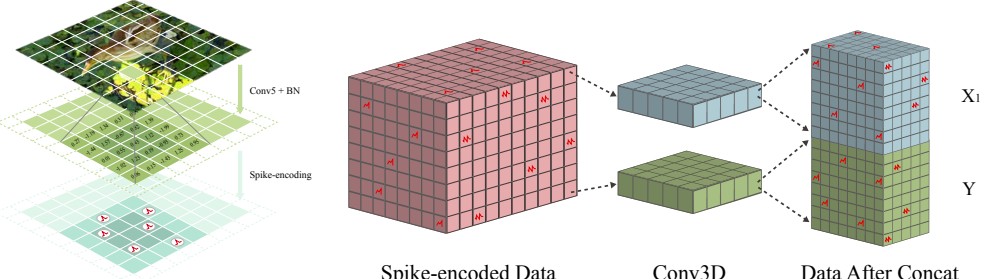

Figure 2: This image contrasts the impact of using different convolutional kernels on spike information encoding. Large convolutional kernels ensure a sufficiently wide receptive field, effectively reducing the information loss caused by spike-encoding and deepening of the model.

Figure 3: This figure illustrates how spike-encoded data undergoes temporal feature extraction. By employing n 3D convolutions concurrently on the temporal dimension of the input matrix and concatenating the resulting feature blocks, the first half of these feature blocks $X_1$ is combined with the input spike-encoded data to form the Guided X input for SGA, while the second half is used as the Y input for subsequent processing within SGA.

## 3.3 SPATIAL-TEMPORAL CORE

In this section, we delve into the core components of our model, which seamlessly integrate SNNs and the Transformer architecture. At the core of our framework lies the Spatial Core, meticulously engineered with three large convolutional kernels. These Spatial Blocks which involve significant convolutional kernels serve a dual purpose: Firstly, they expand the receptive field, allowing for the extraction of essential spatial features. Secondly, each Spatial Block within the Spatial Core is equipped with Spiking Neurons to ensure the spike-encoded transformation of input RGB images. This strategic combination of large convolutional kernels and Spiking Neurons collectively facilitates the comprehensive understanding of intricate spatial patterns and structures present in the input data. In doing so, our model effectively bridges the gap between traditional image data and the spike-based neural processing paradigm. Figure 2 illustrates the transformation of the images from their initial state to the processed state.

Concurrently, the Temporal Core operates at an equally advanced level. As shown in Figure 3, the Temporal Core employs state-of-the-art 3D convolutional operations to extract essential time-domain features from spike-encoded data, as previously described. This intricate process, which involves the fusion of $X_1$ and spike-encoded data $X_2$ (processed through the Spatial Core), facilitates precise analysis and interpretation of temporal dynamics within the data. It provides invaluable insights into the temporal evolution of the visual content and effectively bridges the gap between spatial and temporal information, allowing our model to seamlessly comprehend both spatial and temporal aspects of the input data.

### 3.4 Spiking Gated Attention

Our approach incorporates a highly advanced attention mechanism known as Spiking Guided Attention (SGA), representing a significant breakthrough in the realm of Spatial-Temporal information processing. In contrast to conventional attention mechanisms, SGA places a strong emphasis on the discernment of intricate Spatial-Temporal patterns within the data.

The core principle of SGA hinges on its utilization of Spatial-Temporal processed features. We achieve this by adding the first half of data (X1) processed through the Temporal Core to the earlier spike-encoded data (X2) processed through the Spatial Core, forming the 'X' component. The remaining half of the data, post-processing by the Temporal Core, becomes 'Y'.

The original feature X2, which undergoes the image-to-spike transformation, takes on the role of the 'query' within the attention mechanism. X1 represents the raw input data in its spiking representation. This separation between X and Y empowers our model to effectively align the Spatial-Temporal context provided by 'ST' with the raw input data 'S', thereby enhancing the attention mechanism's capacity to discern and emphasize salient Spatial-Temporal features.

### 3.5 Analyse Of Energy Consumption

In ANNs, a substantial portion of computational load is attributed to Floating-point Operations (FLOPs), with the majority arising from Multiply and Accumulate (MAC) operations. The inherent advantage of SNNs lies in the presence of Accumulate (AC) operations, which serve to reduce the prevalence of MAC operations. Consequently, this reduction in MAC operations leads to a decrease in both FLOPs and energy consumption, aligning with the energy-efficient nature of SNNs.

However, it's worth noting that MAC operations still persist in the data processing pipeline for converting raw images into spike-encoded representations. Therefore, we can estimate the energy consumption by quantifying the number of MAC and AC operations within the computational framework.

The total energy consumption ($E$) can be expressed as follows:

$$E = E_{MAC} \times FL_{conv} + E_{AC} \times (FL_{STCore} + FL_{SGA} + FL_{STMLP}) \tag{13}$$

Here, $E_{MAC}$ and $E_{AC}$ represent the energy consumption associated with MAC and AC operations, respectively. Experimental measurements have determined $E_{MAC}$ to be approximately 4.6 picojoules (pJ), and $E_{AC}$ to be approximately 0.9 picojoules (pJ), based on testing conducted on 45nm technology Horowitz (2014).

This energy consumption calculation enables a precise assessment of the computational and energy efficiency gains achieved by our framework, taking into account both MAC and AC operations throughout the processing pipeline.

## 4 Experiments

We test performance on non-neuromorphic datasets such as CIFAR10 CIFAR100 as well as neuromorphic datasets CIFAR10-DVS, DVS128 Gesture.

Table 1: Comparison with previous works on two neuromorphic datasets, CIFAR10-DVS and DVS128 Gesture. Note that 83.1 % and 98.61 % of achieve the state-of-the-art performance of CIFAR10-DVS and DVS128 Gesture in directly trained SNNs, respectively.

| Method | work | CIFAR10-DVS | | DVS128 | |
|---|---|---|---|---|---|
| | | time-step | Acc | time-step | Acc |
| ANN2SNN | Converted SNN Kugele et al. (2020) | 60 | 65.61 | 60 | 96.97 |
| Direct Training | TDBN Zheng et al. (2021) | 10 | 67.8 | 40 | 96.9 |
| Direct Training | PLIF Fang et al. (2021b) | 20 | 74.8 | 20 | 97.6 |
| Direct Training | SEW-ResNet Fang et al. (2021a) | 16 | 74.4 | 16 | 97.9 |
| Direct Training | MLF Feng et al. (2022) | - | 70.36 | - | 97.29 |
| Direct Training | Spikformer Zhou et al. (2022) | 16 | 80.9 | 16 | 98.3 |
| Direct Training | spike-driven Yao et al. (2023) | 16 | 80.0 | 16 | 97.2 |
| Direct Training | This Work | 16 | **83.1** | 16 | **98.61** |

## 4.1 NEUROMORPHIC DATASETS CLASSIFICATION

**Training.** We keep the same training parameters as the baseline, and the baseline data in the table are our reproduced results. To avoid unfairness, we introduce models from different architectures (VGG, ResNet, Transformer) for comparison, and also include the newer ANN2SNN method in the comparison. In particular, we set the time-steps equal 4 when training the neural dataset. Details of the training hyperparameters and the configuration of each model architecture can be found in the Appendix A.1.

**Results and Analysis.** The experimental results are compared with previous work as shown in Table 1. On the CIFAR10-DVS dataset, our work achieves 82.9% Top-1 classification accuracy, which is a 2.4% improvement over the baseline, and has reached the state-of-the-art in the field of direct training of SNNs. Compared to our baseline implementation, the top-1 classification accuracy on DVS128 Gesture is improved by 1.05%The experimental results on DVS128 Gesture are 98.61% on par with the current state-of-the-art model. Note that only a few of the ANN2SNN data are listed in the Table 1 because of the relatively small number of jobs measured simultaneously at CIFAR10-DVS and DVS128 Gesture.

## 4.2 NON-NEUROMORPHIC DATASETS CLASSIFICATION

**Training.** We keep the same hyperparameters (e.g. BatchSize, LearingRate) as in baseline; the only difference between the two is the network architecture, and our experiments were conducted with the addition of Spatial Core and Temporal Core. The experiments were carried out on an NVIDIA GeForce RTX 3090 device with the architecture STSFormer-4-384-E400. The spikingformer model corresponding to the addition of Spatial Core and Temporal Core was trained with the hyperparameter configuration T=4, dim=384, epoch=400.

**Results and Analysis.** As shown in Table 2 Our model also performs well on non-neuromorphic datasets. On CIFAR10, the model with the addition of Spatial Core and Temporal Core achieves an accuracy of 96.35%, an increase of 0.84% over the widely used Spikformer architecture and 0.4% over the baseline. On CIFAR100, the model with the addition of Spatial Core and Temporal Core achieved an accuracy of 81.39%, an increase of 3.18% over the 78.21% of Spikformer and an increase of 1.04% over the baseline. For the sake of fairness, we also list multiple architectures with multiple comparison methods. As you can see from the table on the CIFAR10 and CIFAR100 datasets, our architecture is way ahead of the curve!

Table 2: Comparison with previous works on CIFAR10 and CIFAR100. Note that 96.35 % and 81.39 % are the state-of-the-art performance of CIFAR10 and CIFAR100 in directly trained spiking neural networks, respectively.

| Method | Work | Architecture | time-step | CIFAR10 Acc | Time Step | CIFAR100 Acc |
|---|---|---|---|---|---|---|
| ANN2SNN | RMP Han et al. (2020) | ResNet-20 | 2048 | 93.63 | 2048 | 67.82 |
| ANN2SNN | Opt Deng & Gu (2021) | RseNet-20 | 512 | 93.58 | 512 | 69.82 |
| ANN2SNN | IP Bu et al. (2022b) | VGG-16 | 16 | 93.38 | 16 | 70.72 |
| Direct Training | HC Rathi et al. (2020) | VGG-11 | 2500 | 92.97 | 2500 | 70.94 |
| Direct Training | BNTT Kim & Panda (2021) | VGG-9 | 20 | 90.3 | 50 | 66.6 |
| Direct Training | Diet-SNN Rathi & Roy (2020) | ResNet-20 | 10/5 | 92.54 | 10/5 | 64.07 |
| Direct Training | TDBN Zheng et al. (2021) | ResNet-19 | 4 | 92.92 | 4 | 70.86 |
| Direct Training | TET Deng et al. (2022) | ResNet-19 | 4 | 94.44 | 4 | 74.47 |
| Direct Training | Spikformer Zhou et al. (2022) | Spikformer-4-384-400E | 4 | 95.51 | 4 | 78.21 |
| Direct Training | spike-driven Yao et al. (2023) | SpikingTransformer-8-768 | 4 | 95.6 | 4 | 78.4 |
| Direct Training | This Work | STSFormer-4-384-400E | 4 | **96.37** | 4 | **81.45** |

## 4.3 ABLATION STUDY

To demonstrate the effectiveness of our proposed ST Core, we test it on the DVS128 Gesture, Cifar10-DVS and CIFAR10/100, and the test results are shown in Tables 7 and 4.

Table 3: Ablation experiments of ST-Core on non-neuronmorphic datasets.

| | CIFAR10 | CIFAR100 |
|---|---|---|
| baseline | 95.95 | 80.37 |
| + Spatial Core | 95.96 | 80.36 |
| + Temporal Core | 96.32 | 81.39 |

As shown in Table 7, we first evaluate the performance of Spikingformer-2-256 as a baseline on two datasets: CIFAR10-DVS/DVS128 Gesture dataset. On the CIFAR10-DVS dataset, we observe improvements in accuracy as we incorporate different components. Specifically, after adding the Spatial Core, the accuracy increases from 80.9% to 81.4%. Further enhancements are achieved by incorporating the Temporal Core, resulting in an accuracy of 83.1%. It's noteworthy that the hybrid model, ST Core, also attains an accuracy of 83.1%. On the DVS128 Gesture dataset, we achieve an impressive accuracy of 98.61% with the incorporation of optimization techniques. We believe that this dataset has been pushed to its limits, and additional optimization efforts have not yielded any noticeable improvements. In summary, our results demonstrate the effectiveness of the proposed Spikingformer-2-256 model, with notable accuracy gains on the CIFAR10-DVS dataset and reaching the limits of performance on the DVS128 Gesture dataset.

In accordance with the guidelines provided in Table 4, we conducted experiments on the CIFAR10/100 datasets to evaluate the performance of our proposed Spikingformer model. As the baseline, we utilized Spikingformer-4-384-400E for testing. After the incorporation of the Spatial Core, we observed a marginal improvement or maintenance of accuracy compared to the baseline.

Table 4: Ablation experiments of ST Core on neuronmorphic datasets.

|  | DVS-CIFAR10 | DVS-Gesture |
|---|---|---|
| baseline | 80.9 | 97.56 |
| + Spatial Core | 81.4 | 98.61 |
| + Temporal Core | 82.9 | 98.61 |

Subsequently, upon adding the Temporal Core, our model exhibited significant accuracy enhancements. Specifically, on CIFAR10, we observed an accuracy increase from 95.95% to 96.32%, and on CIFAR100, the accuracy improved from 80.37% to 81.39%. The most notable results were achieved when both the Spatial Core and Temporal Core were utilized in tandem, yielding the optimal performance. On CIFAR10, the accuracy reached 96.35%, while on CIFAR100, it remained at 81.39%. We would like to highlight that the inclusion of an identical connection mechanism was instrumental in ensuring that the fused ST Core accuracy remained at least on par with the accuracy achieved when the Cores were used separately. These results establish our model as the current state-of-the-art in the respective tasks.

Table 5: Ablation experiments of ST-Core and SGA on neuronmorphic datasets.

|  | DVS-CIFAR10 | DVS-Gesture | CIFAR10 | CIFAR100 |
|---|---|---|---|---|
| baseline | 80.9 | 97.56 | 95.95 | 80.37 |
| w/o ST-Core | 82.1 | 98.61 | 96.04 | 81.09 |
| w/o SGA | 82.9 | 98.61 | 96.35 | 81.39 |
| STFormer (ST-Core + SGA) | **83.1** | **98.61** | **96.37** | **81.45** |

The Table 5 presents results from ablation experiments conducted on neuromorphic datasets to evaluate the impact of different components within our model. The baseline configuration demonstrates initial performance on CIFAR10-DVS, DVS128 Gesture, CIFAR10, and CIFAR100 datasets. When the ST-Core component is removed ('w/o ST-Core'), there is a slight improvement or maintenance of accuracy in comparison to the baseline. On the other hand, when the SGA component is removed ('w/o SGA'), we observe further accuracy enhancements. Remarkably, the best results are achieved when both the ST-Core and SGA components are utilized in tandem, leading to optimal performance. These experiments establish the crucial roles of ST-Core and SGA components in enhancing our model's performance on neuromorphic datasets.

We also experimented with different convolutional kernels for Spatial, and as shown in the Appendix A.2, the best results were obtained at kernelsize=5. This is due to the fact that too large a convolutional kernel causes the model to ignore features of smaller objects.

## 5 CONLCUSION

In this study, we introduced the **ST Core** and **SGA** module to effectively handle spatial and temporal information in image classification. Using a Spike-formal Transformer architecture, we extensively evaluated our approach on CIFAR10 (96.37%), CIFAR100 (81.45%), CIFAR10-DVS (83.1%), and DVS128 Gesture (98.61%). Our experiments have pushed performance limits, achieving state-of-the-art results.

In conclusion, our innovative approach effectively handles spatial and temporal information, leading to state-of-the-art results across various computer vision datasets. This work serves as a foundation for future research and practical applications in addressing a wider range of visual tasks and challenges, with the potential to inspire progress and innovation in the field of SNNs.

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

# A APPENDIX

## A.1 EXTRA EXPERIMENT SETTINGS

In this appendix, we provide essential details regarding the hyperparameter configurations employed in our model, which are pivotal for our experiments conducted across four distinct datasets. The choice of hyperparameters plays a crucial role in determining the performance and robustness of deep learning models. Through careful hyperparameter tuning, we achieve optimal model performance while ensuring its generalizability across various datasets and tasks. The table below summarizes the key hyperparameters utilized in our experiments across the four datasets, serving as a reference point for further research and providing valuable insights into our model's adaptability under diverse conditions.

Table 6: Hyperparameters

| dataset | epochs | time-steps | batch_size | lr&min_lr | opt | sched | weight_decay | amp |
|---------|--------|-----------|-----------|-----------|-----|-------|-------------|-----|
| cifar10 | 400 | 4 | 64 | 5e-4&1e-5 | adamw | cosine | 0.06 | TRUE |

## A.2 ABLATION EXPERIMENT OF KERNEL SIZE

We also conducted ablation experiments on the size of the convolutional kernels. Our findings indicate that the maximum accuracy was achieved when using a kernel size of 5.

Table 7: Ablation experiments of ST-Core on non-neuronmorphic datasets.

| | CIFAR10 | CIFAR100 |
|---|---------|----------|
| baseline (conv3) | 95.95 | 80.37 |
| conv5 | **96.45** | **81.45** |
| conv7 | 92.32 | 78.89 |
| conv11 | 90.85 | 73.15 |
| conv31 | 88.56 | 70.35 |

