# OpenReview forum: "STFormer : Spatial Temporal Spiking Transformer"
_ICLR.cc/2024/Conference — ICLR 2024 Conference Withdrawn Submission_

### Official Review · Reviewer_Yvtr · 2023-10-30

**Soundness:** 2 fair
**Presentation:** 1 poor
**Contribution:** 1 poor
**Rating:** 3
**Confidence:** 5

**Summary:**

This paper proposes a spiking transformer that uses both Spatial Core and Temporal Core to extract spatial-temporal dynamics and designs the Spiking Guided Attention module to fuse them for enhanced contextual understanding. It achieves state-of-the-art results on CIFAR10 and CIFAR100 datasets.

**Strengths:**

1. This paper proposes an architecture that combines temporal and spatial convolutions and introduces a cross-attention-like mechanism to extract richer spatial-temporal features.

2. The proposed method demonstrates significant performance improvements on CIFAR datasets compared to the prior state-of-the-art.

**Weaknesses:**

1. This paper lacks rigor in explaining key technical details. The core modules ST-Core and SGA are only briefly described through figures, lacking precise mathematical definitions or details on the construction methodology. This makes it difficult to fully comprehend the technical contributions.

2. This paper does not provide comparisons to existing spiking transformer architectures. The innovations and contributions compared to prior works are unclear.

3. Due to the unclear details, the credibility of the results is difficult to ascertain. Providing codes or more comprehensive analyses would strengthen the empirical evaluations.

**Questions:**

Major
1. The STFormer seems to sequentially leverages convolutions in spatial and temporal dimensions, and fuses their outputs via a cross-attention mechanism. Please confirm if this understanding is accurate.

2. Figure 2 is the sole explanation of the whole architecture, but lacks legends to clarify components like “CTRL”. More descriptive captions could improve the clarity.

3. The formulations in Eq. 4-12 introduce many new concepts without definitions in the preceding sections. For example, the definitions of SpatialCore and TemporalCore are unclear; it is unknown whether SpikeNeuron refers to LIF; it is still unclear what RPE refers to after reading the full paper, etc.

4. The definition of the Spatial-Temporal Core in Section 3.3 is slightly vague. My interpretation is that the Spatial Core refers to the 3-layer spatial convolution with a residual connection, while the Temporal Core performs temporal convolution after collecting all output spikes in the simulation. Please confirm if this understanding is correct. Besides, the mechanism of the Temporal Core seems to introduce serious waiting latency in actual SNN deployment.

5. It seems SGA alters self-attention to cross-attention for fusing temporal representations?

6. Section 3.5 provides the energy calculation methods, but does not give estimates of the energy consumption on specific tasks. Also, AC operations are closely related to network sparsity during runtime, which is not even mentioned. More details are needed.

7. Testing on larger datasets beyond CIFAR could better demonstrate scalability and applicability to more complex tasks.

Minor
1. Last sentence of Section 2.2 Related Work: "they rarely consider the spike data", the meaning of spike data is unclear.

2. Eq. in the last part of Section 3.1 does not have a correct reference.

3. References 2 and 3 are duplicated.

---

### Official Review · Reviewer_ALKt · 2023-10-30

**Soundness:** 3 good
**Presentation:** 3 good
**Contribution:** 3 good
**Rating:** 3
**Confidence:** 5

**Summary:**

This article suggests Spatial Core and Temporal Core and Spiking Gated Attention (SGA), the resulted spike transformer could achieve state-of-the-art accuracy across almost all dynamic and static datasets. Specifically the writer is able to demonstrate the ablation experiments showing the ST-Core and SGA indeed could boost the performance.

**Strengths:**

see the summary

**Weaknesses:**

1. What is the baseline model on the ablation experiments? Is the baseline model for your own architecture or other study’s baseline?  The study has shown that without the STCore and SGA, the trained model already has excellent performance (80.9% on DVS-CIFAR10) while general accuracy from other studies as shown in your table are below 80%. Would this also mean that the work in accuracy boost may not be very effective (1-2% boost) while spending extra computing resources?

2. Minor mistake on 3.1 preliminaries. Equation 3 referencing.

3. Excellent drawing on figures. However, fonts could be larger fig 1. The words in grey box may be larger.  V_mem, Th_i, U_i^t  too small. “CTRL” long form explanation. Also, font in figure 2 is too small. (Conv5 +BN)

4. Lack of details comparison, such as epochs and number of params, with other state-of-the-art Transformer design. A “table” manner may better emphases the data for readers to justify the improved accuracy is not because of brute-force parameters increment.

**Questions:**

see the weakness

---

### Official Review · Reviewer_Q8zi · 2023-10-30

**Soundness:** 2 fair
**Presentation:** 1 poor
**Contribution:** 2 fair
**Rating:** 3
**Confidence:** 3

**Summary:**

: The manuscript proposes a spiking neural network which combines spatiotemporal preprocessing with a spike-based attention mechanism in order to more comprehensively map transformer-like architectures to SNNs. The resulting performance tables suggest that the approach is more successful than previous literature.

**Strengths:**

The proposed architecture, from a broad-level description, seems like a plausible and appropriate step for neurormorphic network development. The reported task performance seems to confirm the choice of architecture, though comparison to non-spiking networks would help better place the overall performance.

**Weaknesses:**

In its current form the methods are missing too many details or too difficult to follow the overall flow of information in the network, much less to enable a true replication of the presented results. I have listed some specific points below. Without clarification of the network dynamics and architecture I cannot say that the paper is technically sound, and therefore cannot positively judge the contribution.

**Questions:**

1.	The equations describing LIF dynamics (1-3) appear to be incorrect. The reset equation is missing completely. Additionally, there doesn’t actually appear to be any temporal dynamics eg: U_t^L depends on the previous layer (U_t^{L-1}) rather than the previous time (U_{t-1}^(L)).
2.	The flow in figure 1 is difficult to follow. Many abbreviations are missing (Eg: SB, TB, SGA) some of these are defined later, but should be added to the legend here. Finally the LIF of the spatial block appears to terminate, while only the MP (MLP?) activity progresses to the rest of the network. Labeling the temporal block (green) within this figure would also be beneficial.
3.	As the SGA is performing spatiotemporal convolution, this implies that spikes are accumulated over time from the output of the temporal block. It’s unclear from the current write up how the concatenation over time interacts with the fact that the SGA itself should be spiking and evolving through time.
4.	There are several additional hyperparameters other than those in the appendix (eg: number of units in the several linear layers, time constant of the LIF units, parameter initializations). Additionally, the “careful hyperparameter tuning” mentioned in the appendix and “meticulously engineered” kernels (S 3.3) are not sufficiently descriptive to enable replication studies.
5.	Training (4.1 and 4.2) is not sufficiently described. Presumably some surrogate gradient and BPTT are being used on an appropriately chosen loss metric, but this is not described at all.
6.	I am particularly interested in the time constant of the leaky units, given the extremely small number of time steps. While other papers (eg: those presented in table 2) have used similarly small timesteps they typically involve additional tricks, such as sub-timestep spiking. Without additional transformations it is surprising that sufficient spikes would travel through the network in only 4 timesteps, especially with any appreciable leak term.

---

### Official Review · Reviewer_LFnH · 2023-11-03

**Soundness:** 2 fair
**Presentation:** 2 fair
**Contribution:** 2 fair
**Rating:** 3
**Confidence:** 4

**Summary:**

This paper proposes a novel SNN-based feature extraction network called STFormer, which aims to capture spatio-temporal features in images. The network consists of a temporal core using 3D convolutions for multi-timestep feature extraction, and a spatial core with large convolutional kernels to expand the receptive field. Additionally, a Spike Guided Attention mechanism is introduced to enhance correlations between spatial and temporal features. Experiments on CIFAR and DVS datasets demonstrate state-of-the-art performance on image classification tasks.

**Strengths:**

1)The network architecture is novel, considering both temporal and spatial information in SNNs.
2)The Spike Guided Attention mechanism can enhance correlations between spatial and temporal features.
3)Achieves state-of-the-art classification performance on multiple datasets.

**Weaknesses:**

1)The paper does not provide computational complexity and parameter statistics, making it hard to judge model efficiency.
2)No ablation study to validate the efficacy of each component.
3)Experiments are done on limited datasets, only CIFAR and DVS.
4)A large gap exists between simulation experiments and the feasibility of deployment on neuromorphic hardware.
5)Although the performance is high, floating point operations are introduced for some convolution operations, which will affect the deployment of neuromorphic hardware.
6)The description is not in place. For example, the full names of STDP and STBP should be given in the introduction.

**Questions:**

1、What are the advantages of using 3D convolutions for extracting temporal features compared to 2D convolutions?
2、How does the Spike Guided Attention mechanism work specifically? What is the design rationale?
3、There is a problem with power consumption calculation. All convolution operations in SGA are not additive convolutions.
4、Why are no detailed experimental hyperparameters provided for CIFAR100 and CIFAR10-DVS?
5、Why not compare the parameters with the previous architecture [1] [2]?
6、Why not extend the transformer to the Imagenet dataset?
7、Why not unify the captions of all charts at the top or bottom?
8、I am confused about the title of this chapter, section 3.4 SPIKING GATED ATTENTION